

# 1 Cryo-seismicity triggered by ice mass discharge through the Antarctic subglacial hydrographic network

Stefania Danesi[1], Simone Salimbeni[1], Alessandra Borghi[1], Stefano Urbini[2] and Massimo Frezzotti[3]

[1]Istituto Nazionale di Geofisica e Vulcanologia, Sezione di Bologna, Bologna, Italy

[2]Istituto Nazionale di Geofisica e Vulcanologia, Sezione di Roma2, Roma, Italy

[3] Department of Science, Roma Tre University, Rome, Italy

*Correspondence to*: Stefania Danesi (stefania.danesi@ingv.it)

**Abstract.** We analyse seismic time series collected during experimental campaigns in the area of the David Glacier, South Victoria Land, Antarctica, between 2003 and 2016. We observe hundreds of repeating seismic events, characterised by highly correlated waveforms (cross-correlation > 0.95), which mainly occur in the floating area between the grounding and the floating line of the ice stream. The joint analysis of seismic occurrences and observed local tide measurements seems to show that seismicity is not triggered by a seasonal, regular, periodic forcing such as the ocean tide, but more likely by transient irregular impulses. We consider possible environmental processes and their impact on the coupling between the glacier flow and the bedrock brittle failure. Our results suggest that clustered and repeated seismic events may be correlated with transient episodes of mass ice discharge (observed by satellite GRACE and GRACE-FO experiments) through the subglacial hydrographic system that originates upstream of the glacier and extends to the grounding zone, lubricating the interface with the bedrock.

## 1 Introduction

In the recent decades, it has become possible to deploy active and passive seismic instrumentation in the remote regions of Antarctic and Greenland. Since then, it has been demonstrated that seismic activity in polar glacial environments is controlled not so much by tectonic forces but rather by strain and stress fields due to the dynamics of the ice sheet, ice caps and large outlet glaciers (Podolskiy and Walter, 2016 and references therein). When huge masses of ice are in motion, the glacier dynamics strongly affect or even control the seismicity within the ice layer and also at the interface with the bedrock (Podolskiy and Walter, 2016) involving a broad range of frequencies of excitation of seismic waves.

It has been observed that episodes of crevassing and ice cracking (Walter et al., 2009; Colgan et al., 2016; Hudson et al. 2020), stick-slip and basal motion (Anandakrishnan and Alley, 1994; Danesi et al., 2007; Creyts et al., 2009; Smith et al., 2015; Röösli et al., 2016), iceberg calving (O'Neel et al. 2010; Bartholomaus et al., 2012; Walter et al. 2013), glacier noise and tremor (Röösli et al., 2014), hydraulic fractures (Hudson et al. 2020), transient subglacial drainage (Winberry et al., 2009), tidal forcing (Zoet et al., 2012), are some of the main processes capable of raising the tensile and shear stress values until the exceeding of thresholds that trigger seismicity. A wide range of frequencies can be excited in the rupture processes, from icequake episodes (with characteristic frequencies of $10^1$-$10^2$ Hz) to tidal modulation (with characteristic frequencies of $10^{-3}$ Hz).

The David Glacier is the largest ice stream of the Northern Victoria Land region, collecting a catchment basin of ~4% of the East Antarctic Ice Sheet (Rignot, 2002) from Dome C and Talos Dome, and feeding the ~100 km long Drygalski Ice Tongue that floats into the Ross Sea. When the glacier crosses the Transantarctic Mountains, its basal topography has a steep





slope change which generates an imposing icefall (∼400 m downhill) and terminates in the so-called David Cauldron, at the
grounding line level (Bindschadler et al., 2011).
For the David Glacier area, the ice speed derived from satellite radar interferometry (MEaSUREs collection, Rignot et al.,
2017) varies in a range between a few tens of m/y and ∼700 m/y from the plateau towards the proximity of the grounding
line, respectively, with strong acceleration in correspondence of the steepest topographic slopes. For its dimensions, its mas-
sive ice flow, its impact on the dynamics of the front polynya, the David Glacier has been studied at length since the early
90s (Frezzotti, 1993) with important outcomes concerning pioneer and revised estimates of local mass balance (Frezzotti,
1997; Rignot et al., 2019), the ice thickness, surface speed and flux (Frezzotti et al., 2000; Wuite et al., 2009; Le Brocq et al.,
2013; Rignot et al., 2017; Moon et al., 2021), the morphology of the bottom (Tabacco et al., 2000), the grounding line defini-
tion (Bindschadler et al., 2011; Stutz et al., 2021) the subglacial lake system (Smith et al. 2009, Lindzey et al., 2019), the sta-
bility of the Drygalski Ice Tongue (Indrigo et al., 2021), the glacier thinning and retreat in recent geological eras (Stutz et al.,

49    2021).

Based on the analyses of NASA's Ice, Cloud and land Elevation Satellite (ICESat) laser altimeter data, Smith et al. (2009)
identified the presence of six subglacial, hydrologically linked, lakes in the region of the David Glacier catchment. In accord
with Pattyn (2008), the lakes would be responsible for detectable ice mass variations due to transient episodes of drainage
and refilling through the subglacial hydrographic network. More recently, Indrigo et al. (2021) have demonstrated the hydro-
logical continuity between the floating ice channelisation below the David Cauldron and the upstream drainage subglacial
network, which would extend for about 400 km within the David Glacier drainage basin (Le Brocq et al., 2013). A similar
speculation was inferred by Moon et al. (2021), who measured the David Glacier velocity changes between 2016 and 2020
with the application of the offset tracking technique to Sentinel-1A SAR images. These Authors suggest that fluctuations and
transient increases in the ice stream flow speed can be attributed to the injection of ice bottom water through a diffuse sub-
glacial hydrological system.
Temporary seismological experiments have evidenced the occurrence of low-energy seismicity in the glacier area (Bannister
and Kennet, 2002; Danesi et al., 2007; Zoet et al., 2012) providing interesting inferences about possible trigger mechanisms.
Bannister and Kennett (2002) advanced the hypothesis that a deep tectonic lineament or, alternatively, possible fractures in
the ice layer were activated; using data from a dedicated local seismic network deployed around the glacier, Danesi et al.
(2007) suggested that the coupling processes between the ice flow and the bedrock activated the repeating brittle failure of
one or more asperities at the ice/bedrock interface with a stick-slip mechanism; observing a significant correlation with data
collected by the remote TAMSEIS (TransAntarctic Mountains Seismic Experiment) network, Zoet et al. (2012) indicated the
modulation of the ocean tide as a likely forcing on repeating seismicity.
It is worth noting that seismological analyses in extreme scenarios such as Antarctica cannot overlook the particular environ-
mental conditions. For example, by studying the occurrence of icequakes recorded near the Princess Elisabeth Antarctica
Station, Frankinet et al. (2021) demonstrated that the detection threshold of seismic events in Antarctica can be influenced by
weather conditions. In particular, the presence of katabatic winds can significantly increase the level of seismic noise, reduc-
ing the ability to detect events.
In this work, we analyze the evolution of the seismicity observed in the area of David Glacier during three measurement
campaigns in a 14-year long period, between November 2003 and February 2016. We observe that the seismicity shows
highly clustered spatial distributions and indicates evidence of a repeated source which, on the other hand, does not reveal
characteristics of continuity or seasonality over the long time that could be unequivocally correlated with regular recurrent
forcing (temperature cycles, oceanic tides). Our results suggest that clustered and repeated seismic events may be correlated



with transient episodes of mass ice discharge (also observed by the Gravity Recovery and Climate Experiment GRACE and
its Follow On GRACE-FO) through the subglacial hydrographic system that originates upstream of the glacier and extends
to the grounding zone, lubricating the interface with the bedrock.

## 2 Methodology

### 2.1 Seismic data collection

We have collected all available seismic data registered in the study area during three Italian temporary experiments which
were run along 2003/04, 2005/06 and 2015/16 austral summer campaigns around the David Glacier. During the first experi-
ment — dated November 2003 - February 2004 and jointly conducted by an Italian/New Zealand team (Danesi et al., 2007)
— 9 temporary broad-band seismic stations were installed on rock outcrops around the ice stream. The Italian team repeated
temporary observations in summer campaigns 2005-06 and 2015-16 by reoccupying only the sites with the best signal-to-
noise ratio (Figure 1). For these two last experiments, the seismic network was composed of 7 and 6 stations respectively.
New Zealand data and metadata are available through IRIS Data Management Center with the ZL network code (stations
equipped with a CMG-40 seismic sensor and an Orion broad-band digitizer). All Italian stations (network code DY) were
equipped with a Trillium T40 seismic sensor and a Reftek130-01 broad-band digitizer, powered by photovoltaic systems.
One of the Italian stations was located at Starr Nunatak (DY.STAR) in November 2003, more than 50 km away from the
glacier, and during the years it has been equipped to become a semi-permanent seismic station able to record for more than 9
months per year continuously. Due to its stability and performances, the station STAR has been operating up to now provid-
ing a quasi-continuous database of daily waveforms from 2003 to 2017 (last update of the database) including data for most
of the winter seasons.
All the data recovered from the experiments were organized in a SeisComp3 structure (Weber et al., 2007), which allowed us
to perform a systematic and coherent analysis for the whole database.
The availability and data volume for each seismic station involved in one or more experiments between 2003 and 2016,
marked by network code, is shown in the right panel of Figure 1 where it is possible to appreciate the number of stations that
recorded simultaneously during each campaign.

### 2.2 New 3D seismic velocity model of the glacier based on Radio Echo Sounding data

To build a more accurate 3D glacier structural model we used all the Radio Echo Sounding (RES) datasets recorded during
the campaigns conducted by the Italian National Program for Research in Antarctica (PNRA, Programma Nazionale di
Ricerca in Antartide). Datasets covering a period spanning from 1995 to 2016 (http://labtel2.rm.ingv.it/) and the last RES
measurement campaign (austral summer 2015/2016), were focused  on the David Glacier area (black lines in Figure 2A) us-
ing a low-frequency version of the INGV "GlacioRadar" instrumentation operating at 12 Mhz. This particular configuration
was used to integrate the lower number of bedrock reflections available in the Cauldron area in datasets collected at 60 MHz
and 150 MHz (Figure 2B).
Reflection travel times were converted into depth using a constant electromagnetic (EM) wave velocity of 168 m$\cdot\mu$s$^{-1}$ while
cross-point analysis showed a difference in ice thickness of less than 20m in 86% of cases.



The 3D bedrock model of the David Glacier has been obtained by subtracting the ice thickness values from the correspond-
ing RAMPDEM2 (Liu et al., 2015) elevation points for all available RES reflection points since 1995. All the elevation data
(surface and bedrock) were gridded using the "IDW" (Inverse to Distance Weighting) method.
Figure 3 shows the overlay between the ice surface (grey) and the bedrock topography as colour-based relief with isolines at
100 m. All maps are reported in UTM58S projection (WGS84).
Finally, we converted the model into a regular 3D matrix for seismological analyses: we first parameterized a volume of
$124x100x23$ km$^3$ over a regular-cell grid (each cell of $0.5x0.5x0.5$ km$^3$) and then we associated ice or bedrock (or both) to
each cell, as a result of the difference between surface and bedrock heights in the corresponding point of the ice thickness
model. For each cell, we accordingly associated the values of P-wave and S-wave velocity.
For ice layers, we used the compressional-wave velocity Vp=3.8 km/s, and the shear-wave velocity Vs=2.0 km/s (Röthlis-
berger, 1972). For the bedrock we used Vp =6.1 km/s from 0 to 9.5 km depth, 6.3 km/s from 10 to 17.5 km, 6.6 km/s from
depth greater than 18 km, and Vp/Vs ratio of 1.73.
Where the grid exceeded the area of the RES survey and direct observations were not available, we merged mean velocity
values extracted from a 1D layered velocity model (Danesi et al. 2007). Given the geographical extension of the spatial dis-
tribution of nodes, the final 3D velocity structure counts 2352303 values.
**2.3 Seismic Data Analysis**
The seismic database available for the three campaigns was used to characterize the evolution of the seismicity around the
David Glacier. During the three summer campaigns, a network of 9, 7 and 6 seismic stations respectively was operating; dur-
ing mid-seasons and many full years, only the station STAR sited in Starr Nunatak was continuously working (Figure 1).
We have tried to obtain a catalog as rich as possible of earthquake locations, following a 4-step scheme: 1) absolute location
using a seismic 1D velocity structure; 2) absolute location using the local 3D velocity structure; 3) relative earthquake relo-
cation with a double-difference approach and definition of clusters of seismicity; 4) seismic catalog enrichment using a
phase-matched filter detection algorithm.
**2.3.1 Absolute location - 1D velocity model**
For summer seasons, when the full network data were available, we performed a standard absolute earthquake location by
using a simple layered 1D earth velocity structure as already proposed in Danesi et al. (2007) which counts one layer of ice
(1.5 km deep) overlapping four crustal layers down to 33 km depth, and a half-space. For each station, we performed an
STA/LTA trigger algorithm (1 and 30 sec) to detect each variation in amplitude exceeding the signal-to-noise ratio (SNR)
equal to 3, after a bandpass filter in the range 0.4-4 Hz. We merged the list of picks for each station allowing the SeisComp3
(Weber et al., 2007) software to locate the events automatically when a minimum of 5 coherent P-picks was found. Each au-
tomatic location was manually revised by an operator who also revised possible failures in the automatic associations. How-
ever, we decided to keep the locations obtained with at least 5 arrival times recorded at least at 3 stations and with RMS less
than 0.5 s.



### 2.3.2 Absolute location - 3D velocity model


After the manual location, the events were then relocated following a probabilistic approach with the NonLinLoc software
(Lomax et al., 2000) using the 3D velocity model which was obtained from the glacio-radar survey as described in Section
149 2.2.

The final catalog of absolute locations included more than 350 weak events (Ml≤1.8) most located along the steep slope at
the entrance of the David Cauldron, where the glacier topography indicates its maximum declivity and ice flux (Figure 4).
In Table S1 and Figure S1 (both in the supplementary material) all the absolute locations obtained with the 3D model and
global statistic of the results are listed and shown.

### 2.3.3 Relative relocation - double-difference approach


Absolute locations obtained by the NonLinLoc inversion were used as input catalog for a relative double-difference location
with the HypoDD algorithm (Waldhauser and Ellsworth, 2000; Waldhauser, 2001). We intended to apply a coherent inver-
sion scheme for all data available between 2003 and 2016. After the pre-processing, more than 1700 P arrival times and
more than 500 S arrival times for 349 linked events occurring between 2003 and 2016 were selected. We defined an inver-
sion weighting scheme with weight = 1 for P-phases and weight = 0.5 for S-phases, residual threshold = 0.5 s, maximum dis-
tance allowed between linked pairs = 1 km, and damping = 80. After relocation, median residual times result lower than 0.1 s
and median spatial errors  lower than 95 m, 110 m and 195 m respectively for the three coordinates x, y, z.
As expected and in accord with previous works (Danesi et al. 2007; Zoet et al. 2012), the centroids of the two largest clusters
(Figure 5A; 42 and 36 events respectively) were located in the David Cauldron at the foot of the icefall. A significant num-
ber of seismic episodes, collected in 8 additional clusters for a total of 61 events, occurred on top of the icefall mainly after
2005 (Table S2 in the supplementary materials).

### 2.3.4 Phase-matched filter detection analysis and catalog enrichment


Once the relative earthquake location catalog was completed, we applied a phase-matched filter detection technique (Cham-
berlain et al., 2017) to retrieve further possible detections and correlated repeating signals that may have escaped the classic
STA/LTA detection search.
We initially considered only the recordings of the 2003-04 campaign. We selected the seismic trace of a master event for
each cluster (Figure 5B) and applied the phase-matched filter technique to find correlated signals recorded at the TRIO sta-
tion which provided continuous data for the search time interval, excellent signal-to-noise ratio and relatively short distance
from cluster centroids. We used the Python-based EQcorrscan package (Chamberlain et al., 2017) on the continuous seismic
signals to perform the multi-parallel, matched-filter detection, and we were able to detect all the P and S chunks and extract
the waveforms significantly similar (cross-correlation coefficient >= 0.7) to the 10 selected master events representing each
cluster; afterward, we applied the Median Absolute Deviation (MAD) method to correlate the waveforms, providing more
than 1500 cross-correlated seismic signals recorded at TRIO station for the  November 2003 - February 2004 time span.
In the left top panel of Figure 6, we show the vertical component of raw seismic signals for 46 events occurring on Julian
day 324/2003 (November 20th, 2003), filtered in the frequency band 0.4-4 Hz. The similarity among the waveforms (cross-
correlation >= 0.95) and signal duration reveals a common source. On the right top panel the superimposition of 1541 cross-



correlated signals recorded at station TRIO between November 2003 and February 2004 is shown, filtered in the frequency
range 0.4-4 Hz where most of the observed seismic energy is contained (lower panels of Figure 6).
To estimate the temporal variation of these events, we replicated the analysis for the detection of repeating earthquakes using
master events recorded at station STAR and we applied the phase-matched filter technique on the continuous data available
over 14 years, extending the time interval to winters and full years when the temporary seismic network was not installed.
After matching the detection lists, we finally achieved a catalog of repeating events (cross-correlation coefficient between
waveforms ≥ 0.7) grouped in 10 clusters, listed in Table 1. In the bottom panel of Figure 5 we show the larger groups of re-
peating events with red circles centred on cluster centroids.

## 3 Results

The distribution of the seismicity in space and time suggests some preliminary considerations (Figure 5 and Table 1): the
David Cauldron hosted the events with the highest similarity threshold, and a very high number of repetitions (Cl_01 and
Cl_04, nearly 1900), but the seismic activity in this area stopped abruptly and definitively after 2004. In the same period of 3
months, a very large cluster was active on the top (Cl_03, 1188 events) and, again, it stopped after 2004.
Two significant clusters of seismicity were continuously active between 2003 and 2016, albeit with fewer events (Cl_06 and
Cl_10, respectively 318 and 225 events). Both were located upstream of the icefall, along the main branch of the David ice
stream.
The remaining clusters (CL02, CL05, CL07, CL09) have numbers of events that can be disregarded in the following.
Two main questions need to be addressed: why the seismicity collapsed in a few tight areas and why the rate of seismicity
decreased after 2004. To unravel these questions, in the following we first evaluated possible quantitative correlations be-
tween the seismicity and the environmental parameters, the local tide modulation (as suggested in previous works by Casula
et al. 2007 and Zoet et al., 2012) and the wind speed (Frankinet et al., 2021) as possible responsible for the increase of seis-
mic noise that could decrease the efficiency of the detection.

### 3.1 Representation of seismicity and tides

We extracted measurements from the database available at the permanent tide gauge observatory operating off the coast of
Terra Nova Bay about 70 km North from Drygalski Ice Tongue, in front of the Italian base Mario Zucchelli Station which
provides the hourly sampled local wave height (https://www.geoscience.scar.org/geodesy/perm_ob/tide/terranova.htm).
As Figure 7 shows, for a few weeks after the network installation in 2003, from Julian day 320 to 355, we observe that the
repeating seismic activity presents not only similar waveforms but also regular inter-event time spacing, which stands around
24.8 ± 4.7 minutes (Figure 7 a and c) in accord with previous observations (Zoet et al., 2012) and with the application of fric-
tion laws as discussed by Zoet et al. (2020).
The evolution of seismicity exhibits an abrupt change after Julian day 355 when the number of events per day significantly
drops and the inter-event time doubles up to 52 ± 15 min. Seismicity located at the top of the David Glacier icefall remains
sustained during the following years; conversely, further repeating occurrences have not been observed in the David Caul-
dron area after 2004, suggesting that the activation of the downstream events was triggered by a transient forcing.





In the following, we provide some estimates of quantitative correlations between the number of seismic occurrences and en-
vironmental constraints such as tidal modulation, local wind speed and ice mass variation, which could control the activation
of the repeating seismic source.

**3.2 Wavelet cross-correlation with tides**

The correlation between the occurrence of seismicity and the variation in tidal height was verified using the cross-wavelet
transform approach which allows to verify the coherence of two signals over time and to identify any common characteristic
frequencies. Specifically, the inter-event time spacings of the 2003-04 seismic clusters were compared to the tide height. The
WaveletComp 1.1 software through the R package was used for this purpose (Rösch and Schmidbauer, 2018).
As the tide heights data have one sample per hour whereas the inter-event time spacings are irregularly distributed over time,
we chose to compute the upper and lower envelope of the tide time-series, because we expected a correlation with the spring
and neap tides instead of higher frequency tide components, according to Figure 7. These envelopes were used to obtain the
tide heights corresponding to the epoch of each event of the seismic clusters by interpolation. Interestingly, the resulting
cross-wavelet power spectrum (Figure 8) implies that the inter-event time during the 5-weeks clustered seismicity is not cor-
related with the tide period. Conversely, a strong correlation between the two time series can be observed just after Julian
day 355 (after December 21st), with a period of 14 days and the smallest phase difference (Figure 8). This result suggests
that the tidal modulation is the most probable forcing of the seismicity after day 355/2003, possibly controlling the clusters
located between the grounding and the floating lines, while the source of the clustered activity before that date should be at-
tributed to a different cause, limited in time and superimposed over the tidal forcing itself.
Indeed, the grounding and the floating lines (Figure 5B) limit the glacier volume most affected by tidal modulation, being
periodically decoupled from the bedrock and rich in water-saturated sediments (till). The periodic injection of water coil
favour the reduction of the basal shear stress and the acceleration of the glacier.

**3.3 Wavelet cross-correlation with wind speed and meteorological parameters**

For the 2003-04 campaign, we extracted the wind speed measurements for the SofiaB weather station, located 35 km up-
stream from the TRIO station (Figure 1), from the open-access database "Antarctic Meteo-Climatological Observatory at
MZS and Victoria Land" (http://www.climantartide.it).
Frankinet et al. (2021) have studied the influence of the wind on the detection of seismic events during the analysis of the
seismicity recorded at the Princess Elisabeth Antarctic station. Their study shows that the wind speed decreases the sensitivi-
ty of the threshold seismic detection in a non-linear way. Actually, any kind of wind could increase the seismic noise back-
ground but velocity higher than 6 knots (almost 12 m/s) could raise the noise up to 42dB lowering in a drastic way the num-
ber of earthquakes that can be detected. In the case under study, and especially for TRIO station, the comparison between the
wind speed, the root mean squared seismic noise signal and the number of seismic detection (Figure 9a) doesn't indicate a
direct and simple relationship as previously postulated. In fact, the maximum number of seismic picks individuated by the
STA/LTA algorithm was registered before December 1st, when the wind speed was low (<10 ktn), the presence of wind
gusts (> 25 ktn) does not seem to clearly affect seismicity in terms of number of occurrences, nor the root mean squared sig-
nal in the frequency band 0.1-4 Hz, i.e. the characteristic frequency band of the upstream events (Danesi et al., 2007). The
number of seismic occurrences in the second half of the 2003/2004 summer campaign is always lower compared to the num-
ber of detections in the first half, whatever wind speed is registered (Figure 9a).



Finally, we extracted the main meteorological parameters recorded at the SofiaB weather station (atmospheric pressure, air
temperature, relative humidity) for the southern summer 2003-2004 and we calculated the wavelet cross-correlation between
inter-event time spacing and each meteorological parameter (Figure 9b). A non-negligible correlation can be noted between
Julian days 370 and 385 (January 2004) for all weather parameters, with a characteristic period of approximately 3 days. It is
reasonable that in the presence of important crevasses, i.e. along the steepest stretches, liquid water percolates to the bottom
of the glacier and lubricates its interface with the bedrock. It is very likely that the area upstream of the icefall was in this
condition in January 2004, when clusters Cl_01 and Cl_03 were particularly active (Figure 5B and Table1).

**3.4 Ice mass variation from GRACE measurements**

The Gravity Recovery And Climate Experiment (GRACE; 2002-2017) and its Follow-On mission (GRACE-FO; 2018-still
working) give important information related to the Earth gravity field changes due to mass variations. The Gravity Informa-
tion Service (GravIS) of the German Research Centre for Geosciences (GFZ) makes available the products derived by the
satellites missions GRACE and GRACE-FO (Dahle and Murboeck, 2019). In particular, the mass changes of the Antarctic
Ice Sheet are provided in terms of a time-series of gridded ice-mass changes per surface area with a spatial resolution of 50
km x 50 km. The David Glacier was identified in these grids inside the AIS_315 basin (Antarctic Ice Sheet_315). GravIS
database  made available the surface mass density and boundary grids for the basin, which allowed the calculation of the cor-
responding variations in glacial mass pixel by pixel, with a resolution higher than the average value which can be directly
downloaded from the GravIS website. The David Glacier catchment corresponds to the 7 aligned pixels represented in red in
the top panel of Figure 10.
A noteworthy variation of ice-mass, up to 0.4 Gt, was recorded between November and December 2003, in correspondence
with the 5-weeks period under investigation, when the highly correlated seismicity was active. Pixels close to the grounding
line have registered an increase of mass, whereas the backward pixels are characterised by mass reduction.
A transient mass discharge of ice, without comparable repetitions over the following 5 years (Figure 10), was detected in the
catchment area of the David Glacier, as reported by Smith et al. (2015). In particular, Smith et al. (2015) provide an estimate
of the volume variation of about 1.1 km³ over the 5-year between November 2003 and March 2008 possibly linked to the
discharge of water from the subglacial lakes D2 and D3 and the resulting re-filling of D1 (Figure 5A), with the volume dis-
placement rate rapidly soaring up in the first 6 months of the indicated period.
Although there are no direct observations of changes in the glacier basal conditions over the 5 weeks under study, both the
GRACE measurements and the evidence provided by Smith et al. (2015) indicate that a significant circulation of water may
have been injected into the subglacial drainage system below the David Glacier in that period.
The clustered seismic signals show variable duration between 4 and 30 sec, and frequency content concentrated in the 1-5Hz
band, compatible with resonance of subglacial fluids and hydraulic transients, as illustrated by Podolskiy and Walter (2016;
Figures 14 and 15 therein). Our analyses confirm that the transient injection of fluids from the top of the ice stream can be
addressed as the main trigger for seismic occurrences observed at the floating zone level, between the grounding and the
floating line.

**4 Discussion**

We have analysed the evolution of seismicity over a non-continuous period of 14 years around the David Glacier. We have
observed two clusters of low-energy seismic events (Cl_01 and Cl_04 in Figure 5B and Table 1) characterised by highly cor-





related waveforms (cross-correlation coefficient $\geq 0.95$) concentrated in a space of ~2 km$^2$ and in a time-range of 5 weeks
between November and December 2003. Such repeating seismicity was recorded neither in the following months nor in the
following years up to the 2015-16 austral summer campaign. Moreover, significant seismicity has occurred continuously
over the 14 years (Cl_03, Cl_06, and Cl_10) along the largest ice stream that feeds the David Cauldron and channels one of
the main branches of the hydrographic network. Our analyses suggest that weather conditions do not significantly affect the
occurrence of seismicity,
It seems reasonable that the tidal modulation represents the main forcing after mid-December 2003 (Figure 8), but the origin
of clusters Cl_01 and Cl_04 is more likely to be found as a transient, local, non-seasonal forcing superimposed on the tide.
Studies around the coupling between glaciers and the rigid and deformable beds suggest that the dominant processes by
which ice advances and moves past asperities on the bedrock are regelation and plastic flow until the ice speed velocity
reaches a threshold depending on environmental factors such as ice rheology and roughness (Hooke, 2005; Zoet et al., 2020)
which activates the seismicity. Zoet et al. (2020) have estimated that basal stick-slip seismicity can be generally triggered
when the sliding speed is between 500 and 2000 m/yr, which is compatible with the David Glacier flow velocity (Moon et al.
2021 and references therein).
For the period 2003-04 under study, the GRACE measurements of ice mass variation show a trend compatible with the injec-
tion of a sizeable amount of water into the hydrographic network beneath the David Glacier. The associated discharge would
account for the increment in basal lubrication, the possible glacier acceleration, and the reduction of the basal shear stress
which would favour the stick-slip seismic mechanism.
Recent aero-geophysical surveys of the David Glacier catchment area (Lindzey et al., 2019) suggest the revision of the sub-
glacial lake locations and even cast doubts about the presence of real lakes, rather preferring the interpretation of a distrib-
uted system of subglacial drainage.
On the other hand, significant fluctuations in the David Glacier flow velocity, characterised by sudden transient increases in
the ice speed (about 5-10%) with no regularity over time, were observed by Moon et al. (2021). The Authors pointed out that
the observed glacier velocities are inclined to increase during the Antarctic summer for at least three years during the 2016-
2020 period probably due to the extension of the summer melt in ice surface and thus the increase of basal sliding.
The area of velocity increase is upstream the David Cauldron and occurs at elevation more than 600 m. AWS (Automated
Weather Station) data, surface and satellite image surveys do not show any significant melting on snow surface at elevation
higher than 600 m in the area of Terra Nova Bay, therefore the glacier velocity increase observed by Moon et al. 2021 should
be attributed to bottom sliding rather than induced by surface melting. Unfortunately during the 2003-2004 period satellite
images acquisition were not frequent enough to survey any surface velocity change.
**5 Conclusions**
The GRACE data relating to the austral summer 2003 reveal a mass transfer from the David Glacier catchment towards the
coast which could be compatible with an emptying/filling of the regional subglacial hydrographic network and with the con-
sequent acceleration of the glacier flow. The basal lubrication conditions are correlated with the seismic occurrences that we
locate in correspondence with the main flows towards the Cauldron.
Whatever the origin or nature of the subglacial liquid water transfer, the fact remains that seismicity appears to be activated
by basal lubrication, according to Zoet et al. 2020.



Recent measurements (Moon et al., 2021) confirm that the dynamics of the David Glacier is actually affected by transient
summer pulsations of the flow velocity, in the proximity of the grounding zone, both upstream and downstream of the ice-
fall, where the most numerous clusters of seismicity were located.
Although the seismic and satellite observations were not contemporary, we can infer that they both report a recurring (at
least seasonally) behaviour in the glacier, which correlates seismic episodes with the occasional injection of liquid water into
the subglacial hydraulic network.

**Data availability**

All raw data can be provided by the corresponding authors upon request.

**Author Contribution**

Author contribution: SD designed the experiments, carried them out, analysed the seismic data and conceived the manu-
script. SS carried out the experiments and analysed seismic data. AB analysed non-seismic time series and GRACE data. SU
performed the glacio-radar survey and analysed radar data. MF reviewed all available data and led the building of the con-
ceptual model. SD prepared the manuscript with contributions from all co-authors.

**Competing interests**

The authors declare that they have no conflict of interest.

**Acknowledgements**

Maps were produced in the QGIS Geographic Information System environment, using the free database Qantarctica pro-
vided by the Norwegian Polar Institute. We thank DISTART University of Bologna (IT) and DIMEC University of Modena
e Reggio (IT) for providing local tide gauge data. Meteorological parameters were obtained from 'Meteo-Climatological Ob-
servatory at MZS and Victoria Land' of PNRA - http://www.climantartide.it. The Gravity Information Service (GravIS, we
thank Ingo Sansgen) of the German Research Centre for Geosciences (GFZ) provided observational data derived by the
satellite missions.

**Financial Support**

This work was supported by the Progetto Nazionale di Ricerca in Antartide (PNRA 2010/A2.09 2013/AZ2.09) funded by the
Italian Ministry of Research (MUR), and by the MACMAP Project funded by Istituto Nazionale di Geofisica e Vulcanologia
(Environment Department).



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





**Figure Caption**

**Figure 1**: On the left, map of the seismic networks in operation between 2003 and 2016 around the David Glacier. Circles and crosses are used for seismic station sites, coloured in agreement with legend. The meteorological station SofiaB and the tide gauge (installed at Mario Zucchelli Station) are plotted with green triangles. The DEM is extracted from RAMP2 Elevation Model (200m; Liu et al. 2015). The plot on the right shows the availability of the waveforms included in the database. DY: Italian network, ZL: New Zealand network. STAR is a semi-permanent Italian seismic station which has been working up to now since 2003.

**Figure 2:** PNRA RES data coverage in the David Glacier area: A) Flight tracks; B) Radargram at 12 MHz frequency obtained in the Cauldron area.

**Figure 3:** David Glacier area - 3D model of RAMPDEM topographic surface (gray) overlaid to the bedrock elevation (color scaled; isolines at 100m) derived by PNRA RES datasets (Vertical exaggeration=16).

**Figure 4:** Map of epicentres obtained with the standard absolute location of events occurring during austral summers 2003-04, 2005-06, 2015-16 when all seismic stations were recording.

**Figure 5:** A) Distribution of seismicity located in this work and its evolution in space and time: 2003-04 austral summer (red crosses), 2005-06 austral summer (orange crosses), 2015-16 austral summer (blue crosses). The map is developed in a QGis environment, the DEM is extracted from RAMP2 Elevation Model (200m; Liu et al. 2015); the ice flow speed vector field is extracted from MEaSUREs (Rignot et al. 2011, Rignot et al. 2017; Mouginot et al. 2011); subglacial lakes are extracted from the compilation of Smith et al. (2009) - named according to the same nomenclature; the subglacial water flux is from Le-Brocq et al., 2013; the grounding (blue) and hydrostatic (green) lines are extracted from ASAID (Bindschadler et al., 2011). The red square marks the area enlarged in panel B. B) a detailed view of the repeating earthquakes: the red circles indicate clusters and the numbered labels indicate the cumulative occurrences for each cluster over 14 years (when larger than 50). Topography and glaciological information as in the general map.

**Figure 6:** Top left) 46 vertical components of raw seismic signals recorded at station TRIO on Julian day 324/2003, filtered in the frequency band 0.4 - 4 Hz. Signals have cross-correlation coefficients greater than 0.95. Top right) superimposition of the vertical components of 1541 correlated events, recorded at station TRIO between November 2003 and January 2004. Bottom panel) Frequency contents of the corresponding events for TRIO station.





**Figure 7:** a) Horizontal axis is time, expressed in days after 01/01/2003. In blue local ocean wave height as measured by tide gauge (primary vertical axis), in red inter-event time spacing between consecutive events in minutes (secondary vertical axis). b) In blue the local ocean wave height, as in the top panel, the green line shows the number of seismic events per day (secondary vertical axis). c) and d) the histograms show the distribution of inter-event time spacings for a reduced set and the full set, respectively. The corresponding Probability Density Functions (PDF) are superimposed, the vertical dark green lines give the mode value.

**Figure 8:** Cross-wavelet power spectrum between the tide heights and the inter-event time spacing of the seismic events as a function of time (horizontal axis is the 2003 Julian day). The coloured scale indicates the cross-wavelet power level at each period. Black arrows indicate the phase shift between the two time-series: when arrows point to the right the time-series are in phase, when arrows point to the left they are in counter phase. Note the logarithmic vertical scale. The white lines delineate the statistically significant areas, at 10% significance level against a white noise null.

**Figure 9:** Panel A) Hourly wind speed recovered from meteorological station SofiaB during the 3 months November 2003-February 2004 (central panel). The comparison between daily noise (upper) in the two main frequencies of seismic events (0.1-4 Hz) and wind (5-15 Hz) and the number of picks obtained by the STA/LTA detection of station TRIO (lower panel) seems to exclude any correlation with the wind speed. Panel B) Cross-wavelet power spectrum between meteorological parameters and the inter-event time spacing of the seismic occurrences. Coloured scales indicate the cross-wavelet power level at each period - note different scales for each plot. Horizontal arrows pointing to the right indicate that the two signals are in phase at the corresponding period. The white lines delineate the statistically significant areas, at 10% significance level against a white noise null as in Figure8.

**Figure 10:** Ice-mass variation in the David Glacier area observed by GRACE. Left panel: the blue line surrounds the region AIS_315 as in the GravIS catalog. The red area corresponds to the 7 cells (pixels) providing data for this study. Right panel: variation in Gton of ice mass for each cell and each year between 2003 and 2016 represented in accord with the colour scale, recorded by GRACE and GRACE-FO.

24




25

577 **Table Caption**

578 **Table 1:** Geographic coordinates of the centroids of seismic clusters and number of annual occurrences.

579

| Cluster | Centroid coordinates | | Number of detections per year | | | | | | | | | | | | | | |
|---|---|---|---|---|---|---|---|---|---|---|---|---|---|---|---|---|---|
| | Lat (deg) | Long (deg) | 2003 | 2004 | 2005 | 2006 | 2007 | 2008 | 2009 | 2010 | 2011 | 2012 | 2013 | 2014 | 2015 | 2016 | Tot |
| Cl_01 | -75.367 | 160.839 | 803 | 853 | 2 | 5 | 3 | 3 | 0 | 0 | 1 | 0 | 0 | 1 | 2 | 0 | **1673** |
| Cl_02 | -75.330 | 160.405 | 13 | 0 | 0 | 0 | 0 | 0 | 0 | 0 | 0 | 0 | 0 | 0 | 0 | 0 | **13** |
| Cl_03 | -75.385 | 160.511 | 678 | 510 | 0 | 0 | 0 | 0 | 0 | 0 | 0 | 0 | 0 | 0 | 0 | 0 | **1188** |
| Cl_04 | -75.367 | 160.850 | 97 | 93 | 0 | 0 | 0 | 0 | 0 | 0 | 0 | 0 | 0 | 0 | 0 | 0 | **190** |
| Cl_05 | -75.306 | 160.333 | 0 | 0 | 1 | 0 | 0 | 0 | 0 | 0 | 0 | 0 | 0 | 0 | 0 | 0 | **1** |
| Cl_06 | -75.331 | 160.399 | 2 | 6 | 16 | 21 | 20 | 25 | 18 | 1 | 21 | 30 | 27 | 14 | 24 | 0 | **225** |
| Cl_07 | -75.289 | 160.836 | 1 | 0 | 0 | 0 | 1 | 0 | 0 | | 0 | 1 | 2 | 1 | 1 | 1 | **8** |
| Cl_08 | -75.236 | 160.693 | 0 | 0 | 1 | 0 | 0 | 0 | 0 | 0 | 0 | 0 | 0 | 0 | 0 | 0 | **1** |
| Cl_09 | -75.292 | 160.760 | 1 | 3 | 7 | 11 | 9 | 2 | 3 | 0 | 3 | 3 | 2 | 5 | 3 | 1 | **53** |
| Cl_10 | -75.387 | 160.434 | 0 | 3 | 14 | 186 | 22 | 17 | 14 | 3 | 18 | 7 | 10 | 15 | 9 | 0 | **318** |
| | | **Tot** | **1595** | **1468** | **41** | **223** | **55** | **47** | **35** | **4** | **43** | **41** | **41** | **36** | **39** | **2** | |

580

26




**FIGURE 1**

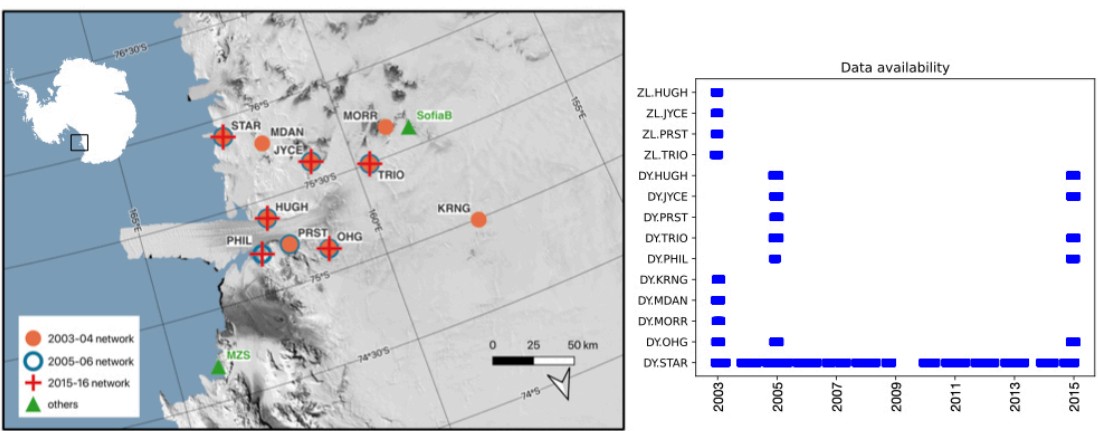

**Figure 1**: On the left, map of the seismic networks in operation between 2003 and 2016 around the David Glacier. Circles and crosses are used for seismic station sites, coloured in agreement with legend. The meteorological station SofiaB and the tide gauge (installed at Mario Zucchelli Station) are plotted with green triangles. The DEM is extracted from RAMP2 Elevation Model (200m; Liu et al. 2015). The plot on the right shows the availability of the waveforms included in the database. DY: Italian network, ZL: New Zealand network. STAR is a semi-permanent Italian seismic station which has been working up to now since 2003.

**FIGURE 2**

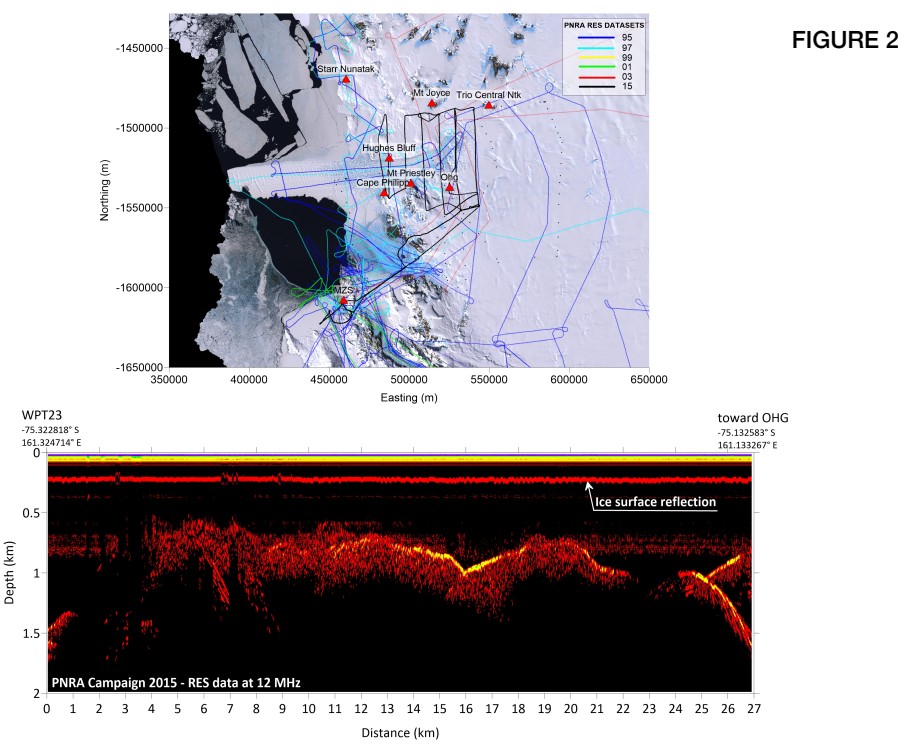

**Figure 2:** PNRA RES data coverage in the David Glacier area: A) Flight tracks; B) Radargram at 12 MHz frequency obtained in the Cauldron area.



**FIGURE 3**

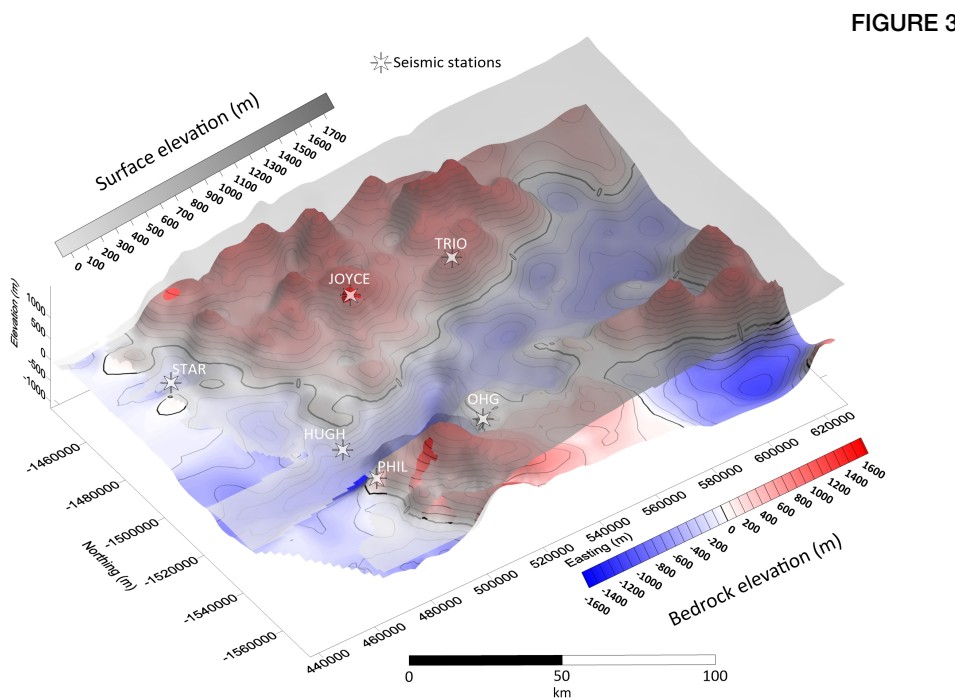

**Figure 3:** David Glacier area - 3D model of RAMPDEM topographic surface (gray) overlaid to the bedrock elevation (color scaled; isolines at 100m) derived by PNRA RES datasets (Vertical exaggeration=16).

**FIGURE 4**

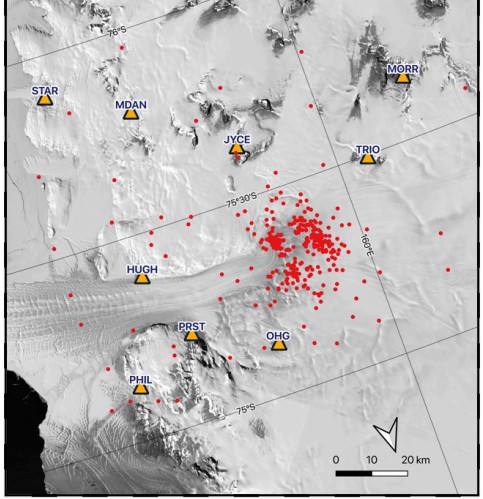

**Figure 4:** Map of epicentres obtained with the standard absolute location of events occurring during austral summers 2003-04, 2005-06, 2015-16 when all seismic stations were recording.



**Figure 5**

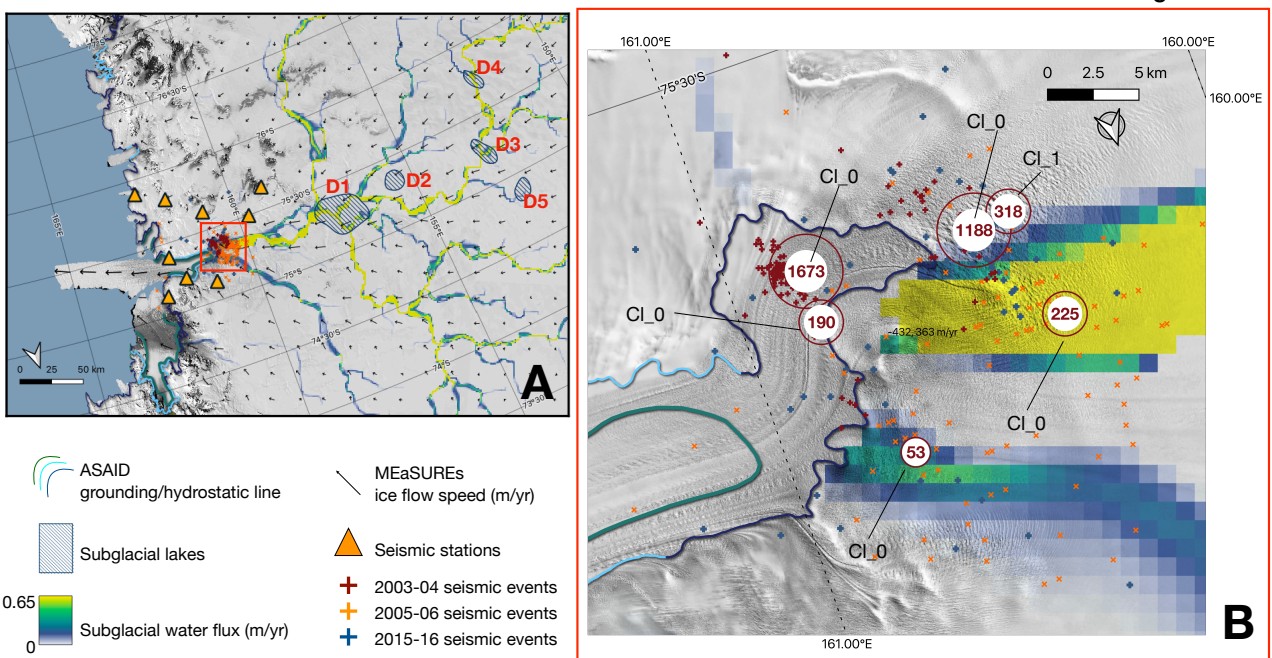

**Figure 5:** A) Distribution of seismicity located in this work and its evolution in space and time: 2003-04 austral summer (red crosses), 2005-06 austral summer (orange crosses), 2015-16 austral summer (blue crosses). The map is developed in a QGis environment, the DEM is extracted from RAMP2 Elevation Model (200m; Liu et al. 2015); the ice flow speed vector field is extracted from MEaSUREs (Rignot et al. 2011, Rignot et al. 2017; Mouginot et al. 2011); subglacial lakes are extracted from the compilation of Smith et al. (2009) - named according to the same nomenclature; the subglacial water flux is from LeBrocq et al., 2013; the grounding (blue) and hydrostatic (green) lines are extracted from ASAID (Bindschadler et al., 2011). The red square marks the area enlarged in panel B.

B) a detailed view of the repeating earthquakes: the red circles indicate clusters and the numbered labels indicate the cumulative occurrences for each cluster over 14 years (when larger than 50). Topography and glaciological information as in the general map.

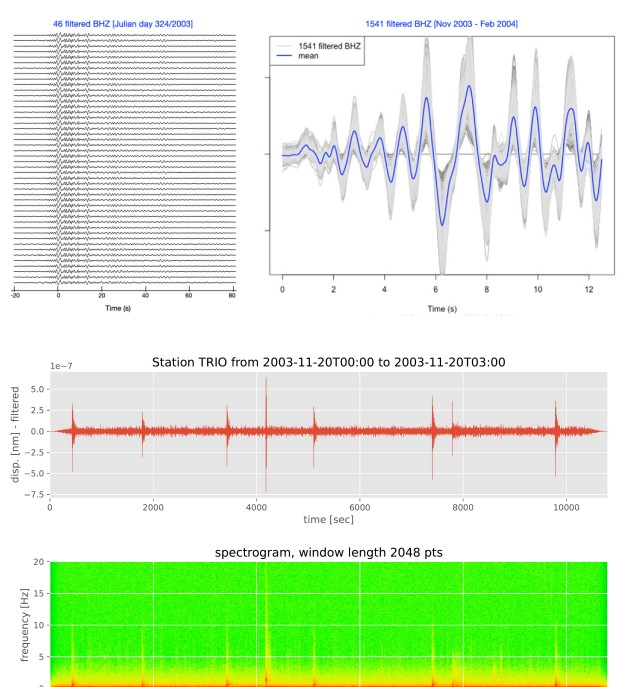

**Figure 6**

**Figure 6:** Top left) 46 vertical components of raw seismic signals recorded at station TRIO on Julian day 324/2003, filtered in the frequency band 0.4 - 4 Hz. Signals have cross-correlation coefficients greater than 0.95. Top right) superimposition of the vertical components of 1541 correlated events, recorded at station TRIO between November 2003 and January 2004. Bottom panel) Frequency contents of the corresponding events for TRIO station.




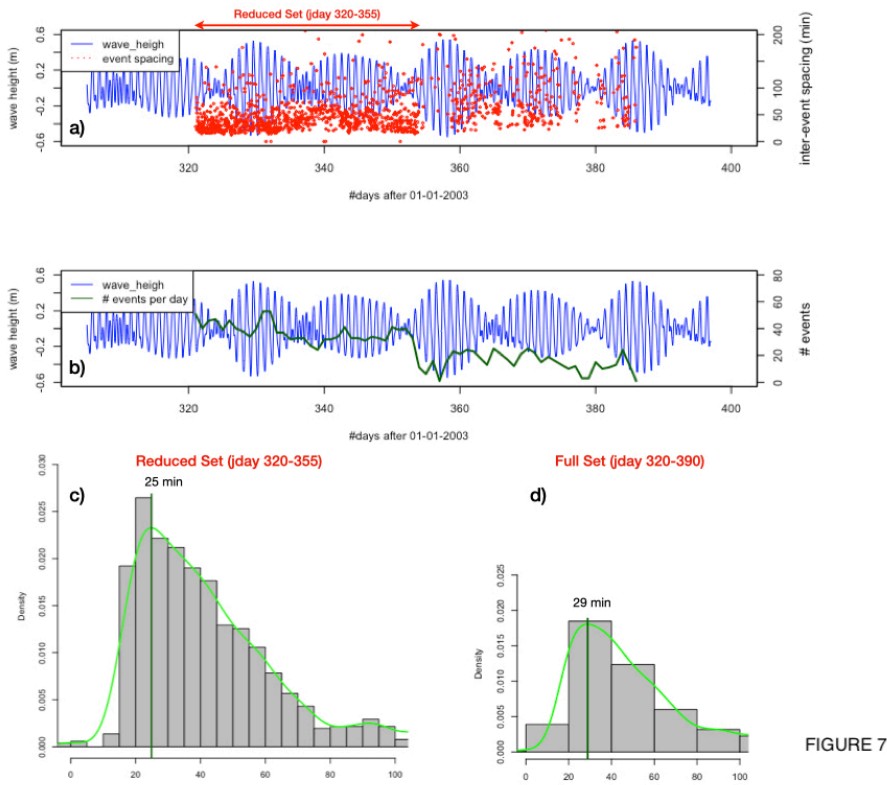

**Figure 7:** a) Horizontal axis is time, expressed in days after 01/01/2003. In blue local ocean wave height as measured by tide gauge (primary vertical axis), in red inter-event time spacing between consecutive events in minutes (secondary vertical axis). b) In blue the local ocean wave height, as in the top panel, the green line shows the number of seismic events per day (secondary vertical axis). c) and d) the histograms show the distribution of inter-event time spacings for a reduced set and the full set, respectively. The corresponding Probability Density Functions (PDF) are superimposed, the vertical dark green lines give the mode value.

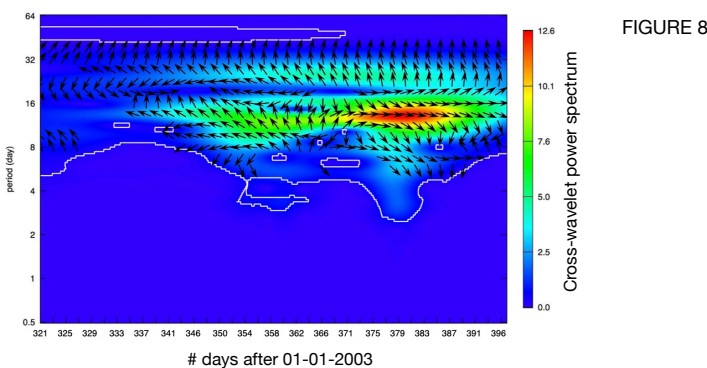

**Figure 8:** Cross-wavelet power spectrum between the tide heights and the inter-event time spacing of the seismic events as a function of time (horizontal axis is the 2003 Julian day). The coloured scale indicates the cross-wavelet power level at each period. Black arrows indicate the phase shift between the two time-series: when arrows point to the right the time-series are in phase, when arrows point to the left they are in counter phase. Note the logarithmic vertical scale.The white lines delineate the statistically significant areas, at 10% significance level against a white noise null.



FIGURE 9

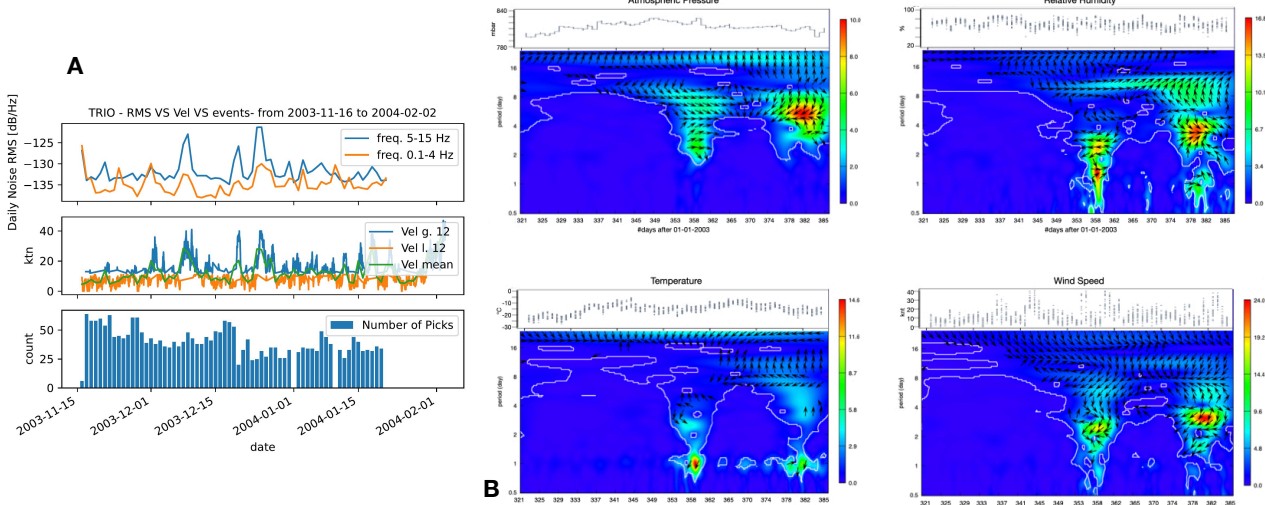

**Figure 9:** Panel A) Hourly wind speed recovered from meteorological station SofiaB during the 3 months November 2003-February 2004 (central panel). The comparison between daily noise (upper) in the two main frequencies of seismic events (0.1-4 Hz) and wind (5-15 Hz) and the number of picks obtained by the STA/ LTA detection of station TRIO (lower panel) seems to exclude any correlation with the wind speed.

Panel B) Cross-wavelet power spectrum between meteorological parameters and the inter-event time spacing of the seismic occurrences. Coloured scales indicate the cross-wavelet power level at each period - note different scales for each plot. Horizontal arrows pointing to the right indicate that the two signals are in phase at the corresponding period. The white lines delineate the statistically significant areas, at 10% significance level against a white noise null as in Figure8.

FIGURE 10

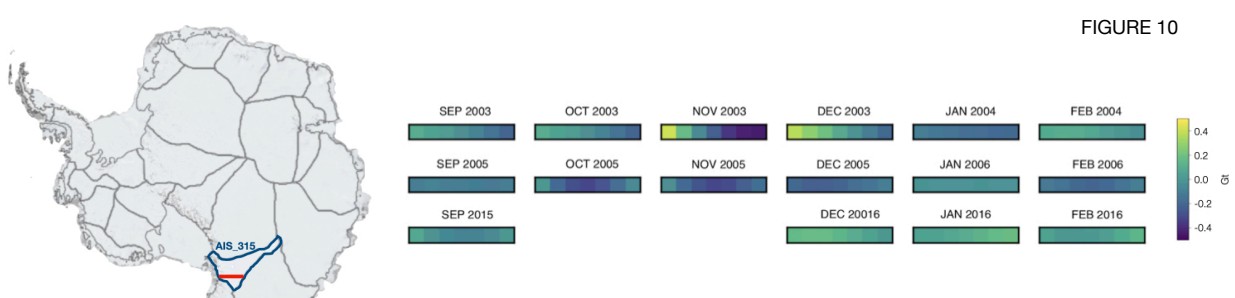

**Figure 10:** Ice-mass variation in the David Glacier area observed by GRACE. Left panel: the blue line surrounds the region AIS_315 as in the GravIS catalog. The red area corresponds to the 7 cells (pixels) providing data for this study. Right panel: variation in Gton of ice mass for each cell and each year between 2003 and 2016 represented in accord with the colour scale, recorded by GRACE and GRACE-FO.