# Peer review of "Cryo-seismicity triggered by ice mass discharge through the Antarctic subglacial hydrographic network"

_EGUsphere, 2022_

## Author Comment (AC3)

Comment on egusphere-2022-29

Fabian Walter (Referee)

MAJOR COMMENTS

The main problem I found with this manuscript is that I could not wrap my brain around the data set. The scientific interpretation seems to hinge on the "ice fall" and the "Cauldron area". Neither of these places is labeled in any of the figures (unless I missed this). In addition, almost all clusters in Figure 5 have the same label. Consequently, when individual clusters or seismic events above or below the ice fall are described, the reader cannot verify or follow the explanations (see specific comments below). The cluster activity and lifetime can also not be verified or put into context.

We agree, thank you. All these points had already been corrected after the first revisions. Figures and manuscript are now corrected accordingly.

A range of dates specifying the data time window is given: 2003/04, 2005/06, 2015/16 (Line 85); 2003-presence (Lines 93-95); 2005-2017 (Line 96); "period 2003-2004" under study (Line 303), 2002-2016 (Figure 1) and perhaps some more. It is not clear which period ends up being the focus and of relevance to this study. This seems to produce some contradictory statements. For example, on Lines 194-196 the authors state that there existed clusters that were active between 2003 and 2016. On Lines 287ff the authors write that repeating seismicity was only observed between November and December 2003. In general, I was not able to understand which events have which characteristics (locations, waveform similarity, clustering behavior, correlation with forcing data, ...).

Time intervals have a uniform format now, and the text has been changed accordingly

If I understand correctly then Zoet et al. (2012 in Nature Geoscience) studied the same glacier and found a forcing by tidal amplitudes. However, the authors of the present manuscript did not attempt to look for this correlation, in fact they purposefully ignore such a correlation against the tidal phase and only investigate the tidal amplitude (Lines 223 ff). Why? It would be interesting to see which events/clusters behave like the Zoet events and which do not. If the authors cannot reproduce the Zoet results, then there may by some problems with the catalogue completeness (even though different seismometers were used for the two studies and the data epochs do not overlap). This deserves discussion.

Actually, Figure 3 in Zoet et al (2012) shows data collected in November 2002, whereas our seismic observations started one year later; this is the reason why we do not try to reproduce their results. However, we do not agree with the statement "in fact they purposefully ignore such a correlation": we have taken into account the hypothesis of tidal forcing seriously, comparing it with the occurrence of seismicity.
The cross-wavelet analysis provides the coherency between the two time-series and estimates the phase between the tidal height and the occurrence of the seismicity as well. The phase differences are reported as black arrows in figures 8 and 9 of the manuscript: rightward arrows mean series in phase and leftward arrows mean out of phase.
Finally, we do not refute the thesis of Zoet et al . (2012); on the contrary, we observe a very good correlation for the events occurring on January 2004 (Figure 8). We are simply

trying to demonstrate that in November 2003 we observed a superimposed transient phenomenon that triggers repeated seismicity, which does not appear to be correlated with the tidal phase.

However, we have tried to clarify better these points adding:

Lines 230-232 state: "This result suggests that the tidal modulation is the most probable forcing of the seismicity after day 355/2003, possibly controlling the clusters located between the grounding and the floating lines, while the source of the clustered activity before that date should be attributed to a different cause, limited in time and superimposed over the tidal forcing itself." And later lines 295-296 : "It seems reasonable that the tidal modulation represents the main forcing after mid-December 2003 (Figure 8), but the origin of clusters Cl_01 and Cl_04 is more likely to be found as a transient, local, non-seasonal forcing superimposed on the tide. "

To be honest, the discussion/conclusion called into question the few take-home messages I thought I had understood when reading the manuscript. First of all, there is a reference to seasonal "behavior" inferred from the seismic data. How is this seasonality backed up in the manuscript? Moreover, the discussion rejects the hypothesis of surface melt affecting basal conditions. Whereas I agree with this, why did the authors come up with this hypothesis in the first place? The temperature data argue against any surface melt. In this regard, it would be interesting to compare the activity of individual clusters that overlapped with the Moon et al. (2021) study to see if there is a correlation with the seasonal speedup.

We have deleted the reference to the seasonality because actually it was not clear.

At line 310 we say: "On the other hand, significant fluctuations in the David Glacier flow velocity, characterised by sudden transient increases in the ice speed (about 5-10%) with no regularity over time, were observed by Moon et al. (2021). The Authors pointed out that the observed glacier velocities are inclined to increase during the Antarctic summer for at least three years during the 2016-2020 period probably due to the extension of the summer melt in ice surface and thus the increase of basal sliding."

This is the opinion expressed by Moon et al - it is not ours: at line 314 we say: "The area of velocity increase is upstream the David Cauldron and occurs at elevation more than 600 m. AWS (Automated Weather Station) data, surface and satellite image surveys do not show any significant melting on snow surface at elevation higher than 600 m in the area of Terra Nova Bay, therefore the glacier velocity increase observed by Moon et al. 2021 should be attributed to bottom sliding rather than induced by surface melting. "

We have added "as suggested by the Authors" to make this point clear.

Unfortunately we do not have overlapping observations because the seismic network was uninstalled at the end of the austral summer, on February 2016. Satellite observations analysed by Moon begin on July 2016.

Source parameters: The authors mention source magnitudes. How were they obtained? This is an important piece of information, because it seems that there is a magnitude overlap with the events by Zoet et al. (2012). Moreover, the size of these events should be discussed and compared to other stick-slip events beneath polar ice streams. It would also be interesting to analyze the waveforms more. Are there different P-phases (refraction through the crust; multiple reflections within the ice column; P-S conversions at

the ice-bed interface; ...)? See next comment. Furthermore, the authors seem to have access to a good azimuthal coverage of recording stations. From this, first motion polarities and focal mechanisms could be obtained that provide relevant information for comparison with ice flow.

As far as the source parameters is concerned, we refer to a previous work (Danesi et al. 2007) that was strictly focused on the seismological analysis of the main clusters of events occurring in 2003 and 2004 (Cl_01 and CL_04). Here we add a significant number of smaller events using a machine-learning technique: being very small they escape the classic detection techniques. In Danesi et al. (2007) we obtained a range of magnitude ML between 1.0 and 1.8 which is very likely an overestimate of magnitude for the events of the present work.
We have evaluated that the first motion polarity technique was not adequate to recalculate the magnitudes for a number of reasons: the number of simultaneously active stations was very low (9 at most, 4-5 on average), we have low energy events (ML<2) and very distant stations (30 - 90 km far), we observe emergent and not impulsive first arrivals, the azimuthal gap is high (median GAP 157 - 167 degrees). For all these reasons, the polarities of the first arrivals would not give robust estimates of the magnitude, especially for the weaker events.
We have added this piece of information in 2.3.2

Locations: Do the locations from the 1-D and 3-D velocity models differ substantially? If so, then this could be related to refracted phases. Moreover, the maps showing event locations have to include error bars. Note that the uncertainties for the relative location is rather low (100 m). How large is the area occupied by location maxima of events within one cluster? Larger or smaller than the error bars? An analysis of the relative locations could give an estimate of asperity size and perhaps even indicate source migration, which would be extremely informative for basal conditions characterizing the material interface at the rupture area (see, e.g., discussion in Gräff et al., 2021, in GRL).

We have changed Figures in order to include errors in the absolute locations. The values of uncertainties in the location are in the table below. The introduction of the 3D velocity model does not improve substantially the deterministic hypo71 location. It should be kept in mind that seismic stations were installed on rocky outcrops, precisely with the intention of observing first arrivals of elastic body waves propagating in bedrock rather than ice.

|  | delta_horiz (km) | delta_Z (km) | median RMS (s) | median GAP (°) | median Minimum Distance DM (km) |
|---|---|---|---|---|---|
| h71_1D | 4.1 | 6.3 | 0.33 | 157 | 28 |
| h71_3D | 4.7 | 6 | 0.34 | 167 | 27 |
| NLL | 8.7 | 11.2 | 0.34 | 166 | 27 |

We are grateful for the reference Gräff et al., 2021, which is a very interesting paper for the modelling of subglacial asperities.The work focuses on small-scale asperities (metric and sub-metric), which is not the scale that our dataset can capture. However, following the referee's guidance, we added an estimate of the extent of the most significant asperities in the David Cauldron.

The text is generally easy to follow, but there are many grammatical errors and phrases, which do not conform to scientific writing standards. It seems that the manuscript was put together in haste, with many orphan paragraphs, false punctuation, single-sentence paragraphs and incorrect terminology. I suggest a thorough re-read and revision from native English speaker. Ok

SPECIFIC COMMENTS

Lines 27ff: Not all the named studies talk about failure mechanisms (e.g., the tremor sources). This has to be reworded.

Yes it is right, we have changed.

Lines 35-49: Pointing out the slope break in the respective figures is extremely important here.

We have changed Figures accordingly with the request.

Lines 50ff: The hydraulic system is indicated in some figures. How was it inferred? This should be discussed better and perhaps be plotted in other figures, too.

We have used the free database Qantarctica as already specified in the Acknowledgments and in the caption of Figure 5. In particular, all the detailed information was already present on lines 528-531 of the manuscript: "the DEM is extracted from RAMP2 Elevation Model (200m; Liu et al. 2015); the ice flow speed vector field is extracted from MEaSUREs (Rignot et al. 2011, Rignot et al. 2017; Mouginot et al. 2011); subglacial lakes are extracted from the compilation of Smith et al. (2009); the subglacial water flux is from LeBrocq et al., 2013; the grounding (blue) and hydrostatic (green) lines are extracted from ASAID (Bindschadler et al., 2011)."

In Section 3.4 we have added some text:
"Subglacial meltwater is largely channeled into the hydrological system beneath the ice sheet, as inferred by Le Brocq et al. (2013) from satellite and airborne remote sensing. As for the David basin, the subglacial drainage network conveys the flow into two main subglacial streams (Figure 5) which feed the outlet glacier."

Line 65: "significant correlation with data" is too unspecific.

It is true, thank you. We have changed "a significant correlation between inter-event spacing to tidal amplitude in data collected by …"

Line 109: "integrate the lower number of bedrock reflections" is unclear.

Yes, ok, changed with "low number". This particular configuration was used in order to improve the bedrock model in the Cauldron area where RES reflections were more difficult to obtain (Figure 2B).

Lines 116-117: This information belongs in the figure caption, not the main text.

Yes, we have moved the text and added to the caption.

Line 123: Does the "9.5 km depth" refer to below the surface or below the ice sole?

This layer thickness is referred to the bedrock, therefore it is below the ice sole. It is specified at line 123.

Lines 125-126: "we merge mean velocity values" is unclear.

Ok, we have rephrased "we merged velocity values extracted from a 1D layered velocity model"

Line 131: Specify "many full years". Changed in 14 full years

Line 142: What are "coherent P-picks"? Changed: "a minimum of 5 P-picks was associated to the same event"

Lines 143-145: Why did you decide this way? We have added "As a precaution"

Line 147: "manual locations": From the explanations above it seemed that the locations were automatic. We have made it clearer

Lines 150-151: Indicate this location on the map. Done

Lines 170ff: You mention what you did "initially", but what did you end up doing? What is the difference between the two correlation scans?

The first scan was performed on 3 months of seismic data recorded at the nearest station (TRIO). Then, we replicated the analysis on the database of data collected over 14 years from the semi-permanent station (STAR). We have changed the text to make it clear.

Lines 199ff and elsewhere: The authors tend to announce what they are about to do in upcoming sections. This is a waste of space.

In this case we do not agree with the Reviewer. In our opinion, these few lines help the reader to follow the line of reasoning

** Lines 211ff: The shift in inter-event times needs to be backed up with a figure.

Ok, done in figure 7 and figure 8

** Line 226: "by interpolation" is too unspecific.

We have specified linear interpolation

** Lines 233ff: This paragraph is unconnected from the rest of the text.

We have moved these lines to the Discussion

** Lines 237ff: The correlation between trigger threshold changes and wind speed can be complicated. It can depend on how you parameterize wind speed, i.e. via the noise floor or peak. The latter would isolate gusts. If a noise floor (e.g., lower percentile), median or mean is chosen then you may miss the influence of gusts. We have added this piece of information in the text. We have specified that from the hourly-spaced time series, we have calculated the daily mean wind speed to perform the cross-correlation. It is visible in Figure 9A as well

Line 247: "December 1st" of which year? Changed to 2003/12/01

**Line 267: "a resolution higher than the average value": Unclear.

Text changed to: "The David Glacier was identified in these grids inside the AIS_315 basin (Antarctic Ice Sheet_315). GravIS database made available the surface mass density and boundary grids for the basin, which allowed the calculation of the corresponding variations in glacial mass pixel by pixel. This passage allowed to improve the resolution of the gravity field to the pixel level, rather than to the basin average value - which can be directly downloaded from the GravIS website."

Lines 273ff: Is Smith et al. (2015) the right reference? It talks about Rutford Ice Stream, not David Glacier. "soaring up" has to be quantified. Changed the reference - and changed in "rapid increment"

** Lines 281ff: A waveform record supporting the fluid resonance is needed. There could be other explanations for dominant frequencies. I do not see how the last sentence in this paragraph follows.

We have added this information in Figure 6, and added the reference in the main text.

Lines 297ff: The discussion about sliding physics seems oversimplified. First of all, I suggest not only referring to regelation and plastic flow. This holds for hard beds, but there are other mechanisms in the context of soft beds. Zoet and Iverson (2020 in Science) should be discussed. Finally, is it really valid to consider a simple velocity threshold for seismogenesis? After all, it seems that large changes in basal hydraulic pressures were involved.

Thank you for this suggestion. We have added "These observations are in agreement with the thesis of Zoet and Iverson (2020) who showed that water-saturated till in soft glacier beds is able to weaken and destabilise the ice-bedrock interface owing to the increment of pore water pressure (Zoet and Iverson, 2020)."

Lines 310ff, including the next paragraph: Ideally, some ice flow velocities should be shown here. Is there absolutely no data from 2003? If so, this statement should be made more concrete (there certainly exist flow velocity estimates) and backed up with references.

In the manuscript we say : " Unfortunately during the 2003-04 period satellite images acquisition were not frequent enough to survey any surface velocity change"

Nowadays satellite images acquisition is frequent enough (ca 36 days) to allow the comparison with the trend variations observed by Moon et al. This was not the case in 2003, when satellite images were acquired at time intervals of once or twice a year and they did not allow the recognition of transient variations over a few tens of days such as those discussed in this article.

We have clarified it in the text.

FIGURES

In general, panels should be labeled a, b, c, ... or equivalently and not be referred to with "upper", "lower", "right", "left", ...

Figure 1: Mark ice fall and Cauldron. Give a date for "now" in caption.

Ok done

Figure 2: Mark Cauldron. Where was the radar transect taken? Annotate/label the radar transect (e.g., fast flowing ice vs. ice sheet).

Done, also accordingly with other referees request

Figure 3: Not sure if showing the surface in grey shade has an added value. Ok

Figure 4: Is this from the 1-D or 3-D inversion? Show error bars. Ok done

Figure 5: How were the grounding/hydrostatic lines inferred? I would explain them in the text, too. How was the subglacial water flux determined? This is important information, by the way! Show error bars. Where are the cluster events? Are they masked by the number circles? This would be unfortunate, actual locations would be informative. Ok changed

Figure 6: Fonts are too small. What are the different grey shades in the top right panel? Ok changed

Figure 7: Fonts are too small. Which clusters do the events shown in the top panel belong to? Vertical green line is missing or hard to see. Ok corrected

Figure 8: How was significance determined/defined?

The WaveletComp R-Package for cross-wavelet analysis performs a statistical test to verify the null hypothesis that there is no periodicity in the considered time-series. The tests are performed with a significance level of 10%. This information has been added in the caption.

Figure 9: Fonts are too small. Ok changed

Figure 10: The horizontal bars should be shown against some along-flow-line quantity like ice thickness or bedrock elevation. A reference for the mass change data should be given in the caption (in view of GRACE).

The reference for the mass change data was already reported in the text. It was added also in the figure caption: "from the Gravity Information Service (GravIS) of the German Research Centre for Geosciences (GFZ)"